# Adherence to Annual Fundus Exams among Chinese Population with Diagnosed Diabetes

**DOI:** 10.3390/jcm11226859

**Published:** 2022-11-21

**Authors:** Yifan Zhou, Xiaowen Li, Qinglei Sun, Jin Wei, Haiyun Liu, Keyan Wang, Jianfeng Luo

**Affiliations:** 1Department of Ophthalmology, Putuo People’s Hospital, Tongji University, Shanghai 200060, China; 2Department of Biostatistics, School of Public Health, Fudan University, Shanghai 200032, China; 3NHC Key Laboratory of Health Technology Assessment, Fudan University, Shanghai 200032, China; 4Key Laboratory of Public Health Safety of Ministry of Education, Fudan University, Shanghai 200032, China; 5Department of Ophthalmology, Shanghai East Hospital, Shanghai 200120, China; 6Department of Ophthalmology, Shanghai General Hospital (Shanghai First People’s Hospital), School of Medicine, Shanghai JiaoTong University, Shanghai 200080, China; 7National Clinical Research Center for Eye Diseases, Shanghai 200080, China; 8Shanghai Key Laboratory of Fundus Diseases, Shanghai 200080, China; 9Department of Ophthalmology, Eye and ENT Hospital of Fudan University, Shanghai 200031, China

**Keywords:** adherence, fundus exams, diabetes, China, CHARLS

## Abstract

Adherence to annual fundus examinations in the Chinese population with diabetes and its correlates have not been investigated. The present study obtained data for the first nationally representative survey in China, China Health and Retirement Longitudinal Survey (CHARLS), which collected a wide range of data every 2 years, including demographic, socioeconomic, medical and lifestyle-related information. The adherence rates to annual fundus exams across four waves (2011–2018) were assessed. Univariate and multivariable logistic regressions were used to determine factors associated with adherence. The adherence rates to annual fundus examinations of ou study population were 23.6% in 2011, 15.3% in 2013, 17.5% in 2015 and 21.5% in 2018, respectively. Consistent results over four waves showed that non-adherent patients had a relatively lower educational level, insufficient diabetes medication use, fewer non-medication treatments and irregular physical examination compared to those who were adherent to the annual fundus exam (all *p* values < 0.05). These variables were further identified as factors associated with adherence according to univariate and multivariate logistic regression analyses (all *p* values < 0.05). The present study provides explicit evidence that the adherence rate to annual fundus examinations among Chinese population with diabetes is worryingly low. Insufficient educational attainment, especially specific diabetes education, has a negative impact on patients’ adherence to clinical guideline for eye health.

## 1. Introduction

Diabetes mellitus (DM) is a major public health problem worldwide [1]. Compared to other systemic diseases, DM more frequently leads to vision-threatening events, such as diabetic retinopathy (DR) [2]. The prevalence of both diabetes and DR is on the rise and it has already become the world’s leading causes of blindness among labor population and senior citizens [3]. Early detection and timely treatment are key strategies for DR interventions and the resulting vision impairment [4]. Proper adherence with screening guidelines for the early detection of DR and DR interventions could reduce the occurrence of severe vision losses by up to 90% [5]. Thus, joint guidelines set by authority organizations recommend annual screenings for patients with diabetes [6,7]. However, the conditions of the normative management of regular fundus examinations among patients with diabetes are not satisfactory in many countries [8,9,10,11,12,13,14,15].

Researchers have reported low adherence rates to annual fundus exams among the population with diagnosed diabetes to date [8,9,10,11,12,13,14,15]. Multiple factors have been considered to impacts on the poorer adherence to regular screenings in the population with diabetes, including racial disparities, coexistent diseases, educational level and insurance coverage [8,9,10,11,12,13,14,15,16]. Population-based data also indicate that a significant proportion of patients with diabetes receiving regular public medical services are suffering from undiagnosed ocular diseases [17,18], which reflects the insufficient acknowledgement and attention attached to the eye-care of patients with diabetes from the whole medical system.

Compared to the relatively higher adherence rate (over 50%) in some developed countries [19,20], according to some recent reports, the significantly lower adherence rates in developing countries are even more worrisome. To date, there are still very few nation-wide reports on adherence to regular fundus examinations among patients with diabetes from developing countries. China is the most populous developing country; it also has the largest number of patients with diabetes. The China Health and Retirement Longitudinal Study (CHARLS) surveyed personal information from a wide range, including demographic, socioeconomic, clinical, health conditions and lifestyle information, to facilitate research on the middle-aged and older Chinese population. Thus, here, for the first time, the present study aims to introduce the conditions of adherence to annual fundus examinations and its associated factors among the Chinese population with diagnosed diabetes.

## 2. Materials and Methods

### 2.1. Patients and Public Involvement

Data were obtained from the China Health and Retirement Longitudinal Study (CHARLS) 2011 (wave 1, baseline), 2013 (wave 2), 2015 (wave 3) and 2018 (wave 4), which is the first nationally representative longitudinal survey sampling residents (middle-aged and older adults, ≥45 years old) from 150 counties across 28 provinces in China. With a response rate over 80%, CHARLS provides the most up-to-date longitudinal datasets for studying the health status and well-being of the middle-aged and elderly population in China.

### 2.2. CHARLS Datasets

Initiated in 2011, CHARLS was performed using a four-stage stratified cluster sampling method. First, 150 county-level units across 28 provinces in China were sampled to represent a mixture of urban and rural areas with a wide variation in terms of economic development. Next, primary sampling units (PSU), administrative villages in rural areas or residential neighborhoods in urban areas, were sampled within each county, resulting in a total number of 450 villages/neighborhoods. Then, the dwellings in each PSU were mapped and 24 of the mapped households in each PSU were sampled for further studies. At every stage, further sampling was performed through random selection. In each selected household, one individual aging 45 or above was invited to participate together with his or her spouse, if available.

### 2.3. Definition of Diabetes and Adherence to Annual Fundus Examinations

In CHARLS, diagnosed diabetes was defined as the situation in which the participants responded ‘yes’ to the questions: ‘Have you ever been diagnosed with diabetes or high blood sugar by a physician?’. According to the recommendations from several authority organizations including American Diabetes Association, American Academy of Ophthalmology, American Association of Clinical Endocrinologists and Chinese Medical Association, patients with diagnosed diabetes in CHARLS were considered adherent if they confirmed having fundus examinations within the past 12 months and non-adherent if they denied receiving such examinations.

### 2.4. Independent Variables

According to previous studies, multiple factors that could potentially affect patients’ adherence to regular fundus examinations were adapted into the statistical models in the present study, including age, gender, marital status [21], educational level [21], insurance [22], multi-morbidities [23], self-reported vision impairment [24,25], drinking status [21], smoking status [21], usage of medication for diabetes [26] and application of non-medication treatments (weight control, physical exercise, diet, smoking control and foot care). Criteria for these variables were described in previous CHARLS studies.

### 2.5. Statistical Methods

In the present study, demographic, socioeconomic, medical and lifestyle-related factors were compared between patients with diabetes who endorsed versus those who denied a fundus examination in the last year using analysis of variance, Wilcoxon test, Pearson χ^2^ or Fisher’s exact test, according to the types and distribution of data. Univariate logistic regressions were conducted to assess the associations between independent variables and adherence to annual fundus exams. Multivariable logistic regressions were then performed using filtered variables (with *p* values < 0.1) from univariate analysis to determine which variables were independently associated with improved adherence to annual fundus exams. Results were reported as odds ratio (OR) with 95% confidence intervals (95% CI), and *p* values < 0.05 were regarded as statistically significant in the present study. A statistical analysis was performed using SAS 9.4 statistical software (SAS Institute, Cary, NC, USA).

## 3. Results

The final analytic sample of the present study was 985 from CHARLS 2011, 1025 from CHARLS 2013, 1027 from CHARLS 2015, and 1050 from CHARLS 2018 (Figure 1). The adherence rates to annual fundus examinations of our study sample were 23.6% in 2011, 15.3% in 2013, 17.5% in 2015 and 21.5% in 2018, respectively (Figure 2). There were statistically significant differences of adherence rates between 2011 and 2018 (*p* value < 0.005, Appendix A).

Characteristic differences between patients with diabetes who were adherent or non-adherent to annual fundus exams are shown in Table 1. There were some consistent results over four waves: non-adherent patients had relatively lower educational level, reduced diabetes medication use and fewer regular physical examinations compared to those who were adherent to annual fundus exams (all *p* values < 0.005). From 2013 to 2018, among patients who denied annual fundus exams, there was a significantly lower proportion of patients having non-medication treatments, compared to those complied with annual fundus exams (all *p* values < 0.005). The descriptive characteristics of patients with diabetes who failed to comply with annual fundus exams from 2011 to 2018 are further exhibited in Appendix A.

The univariate logistic regression analysis revealed the potentially associated factors of adherence in our sample (Table 2). There were consistent results, showing that factors such as higher educational level, using diabetes medication, application of non-medication treatments and adherence to regular physical exams were significantly and positively correlated to patients’ adherence to annual fundus exams (all *p* values < 0.05). According to multivariate logistic regression analysis, certain variables, including higher educational level, using medication or non-medication treatments, and adherence to regular physical exams, were also shown to have explicit and profound correlations with adherence to annual fundus exams (Table 3, all *p* values < 0.05).

## 4. Discussion

The latest global research on causes of blindness and its trends over 30 years indicated that DR is the only vision-threatening disease with increased prevalence between 1990 and 2020 [27]. To reduce vision impairment and consequent disability associated with DR, it is crucial to investigate the situation of adherence to regular fundus examinations, which is a secondary level of prevention, and to identify modifiable risk factors of patients’ adherence. To the best of our knowledge, the present study is the first nation-wide report on this issue from the Chinese population with diabetes.

Overall, the adherence rates of annual fundus examinations according to CHARLS were 23.6% in 2011, 15.4% in 2013, 17.2% in 2015 and 21.5% in 2018, respectively (Figure 2). Such adherence rates were relatively lower than those reported in certain studies performed in some developed countries, such as the United States of America [15,20], Korea [28] and Germany [19], which indicates the worrying condition of this issue in our country. There was a recent report of an adherence rate of 33.3% among Chinese patients with diabetes [29]. However, that research was a clinic-based study, which focused only on a small sample of patients from Guangdong province.

Above all, we proposed that the low adherence rate might be due to the low self-awareness rate of diabetes among the Chinese population. The stubbornly high prevalence of undiagnosed diabetes is a major problem in diabetes prevention and control in both developed and developing countries [30]. According to IDF diabetes Atlas, the self-awareness rate of diabetes in China was probably less than 50% in the last decade [31]. Such a condition was echoed in CHARLS. There are two main definitions of diabetes in CHARLS: self-reported diabetes and reference-defined diabetes, which have been described in the CHARLS protocol [32,33]. Blood tests could contribute to a more accurate estimation of the self-awareness rate. The self-awareness rate regarding diabetes among our study sample in 2011 baseline was 46% (Appendix A). This self-awareness rate is consistent with other CHRALS studies [33]. We further analyzed the adherence rate among people who were not aware of their diabetes (diabetes identified by blood test upon survey, but the patient responded ‘no diabetes’ in the questionnaire). Not surprisingly, but appallingly, none of these patients had attended any fundus screening in the last 12 months (Appendix A). Thus, the realistic adherence rate among our population might be greatly lower than that assessed in the present study. Hence, we propose that one important intervention to promote adherence to regular fundus examinations among the population with diabetes is to reduce the rate of undiagnosed diabetes in our population as a precondition in China.

Non-adherent patients had a significantly lower educational level, reduced diabetes medication use or less application of non-medication treatments for blood glucose control, and fewer regular physical examinations compared to those who were adherent with annual fundus exams in the present study (Table 1). Consistently, according to further logistic regression analyses, these variables were also identified as factors that could possibly affect the adherence to annual fundus exams among our study sample (Table 2 and Table 3). We propose that all these significantly associated factors could be correlated with each other in some way or another. We hypothesized that the attainment of education could play a central role.

Education level has been reported to have close a relationship with patients’ self-consciousness and knowledge of diabetes [34,35]. As mentioned above, improvements in self-consciousness of diabetes should come prior to the promotion of adherence to regular fundus examinations among our population with diabetes. From a specifically ophthalmic level, lower educational attainment may limit patients’ knowledge of vision-threatening ocular diseases [36,37,38]. Less educated people are less likely to be exposed to educational materials and may find it harder to acquire essential information from educational materials [38]. They may, therefore, lack knowledge of the rationale behind the multiple complications caused by diabetes [39], and consequently, could not understand the importance of normative management of diabetes, including periodic fundus exams [38,39]. Thus, insufficient knowledge about diabetes has been raised as the most significant barrier to eye-care among the population with diabetes [11,38,40], and lack of specific diabetes education has been considered the main reason for non-adherence to annual fundus exams among patients with diabetes [8,9,10,13], which may have detrimental impacts on their the self-management of diabetes, including taking diabetes medication or non-medication treatments and receiving regular physical examinations [34,35,38,41,42,43].

Some other factors, such as insurance coverage, vision impairment, and multi-morbidities, have been reported to have certain associations with adherence to the annual fundus exam among patients with diabetes in previous research. However, we did not notice consistent and profound evidence in the current study.

The present study is the first to introduce a nation-wide condition of adherence to regular fundus examinations among the population with diabetes in China, which included a large sample size from a nationally representative population. A wide range of factors, including demographic, socioeconomic, medical and lifestyle-related information, were adapted in statistical models. Meanwhile, we acknowledge several limitations to our study. First, most data from CHARLS were self-reported and might be subjected to recall bias or inaccurate identifications of medical conditions. Some factors, such as physician visits and glycosylated hemoglobin, were not included in our statistical models, due to the inaccessibility of CHARLS datasets. We also lacked PSU data in CHARLS 2015 and 2018; therefore, the adherence rate of our study sample cannot be seen as the adherence rate of the whole country.

## 5. Conclusions

The adherence rate to annual fundus exams among patients with diabetes in China is worryingly low. This situation should receive immediate attention. Insufficient educational attainment is found to be the main reason for non-adherence to periodic eye exams among the Chinese population with diabetes in the last decade. Especially, specific diabetes education and the consequent normative managements of diabetic condition could be addressed as efficient interventions that are supposed to have great impacts on patients’ adherence to clinical guidelines for eye health and the prognosis of diabetes-related complications.

## Figures and Tables

**Figure 1 jcm-11-06859-f001:**
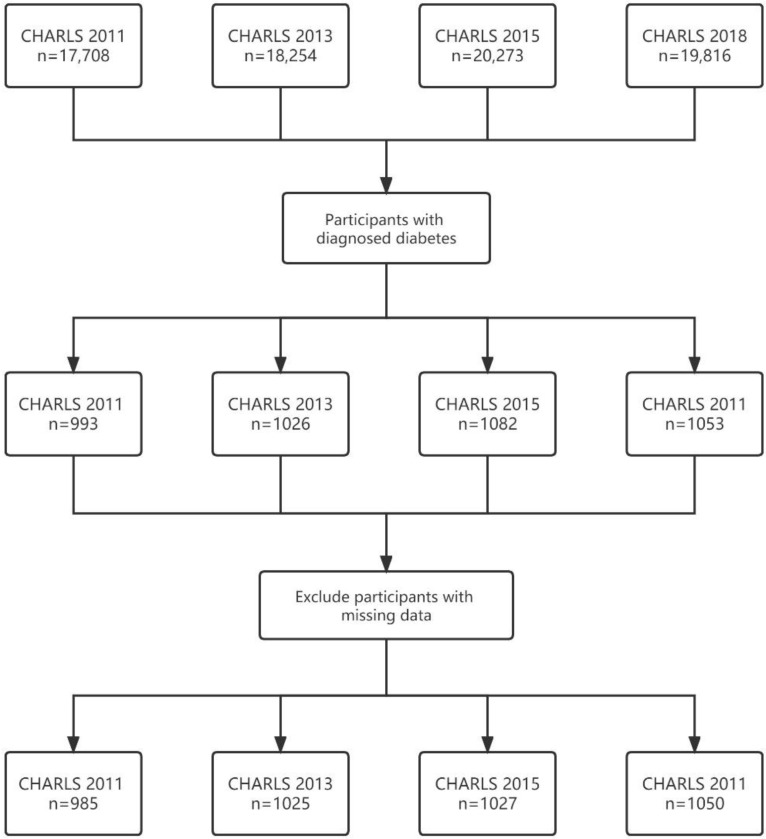
Sample screening of diabetic participants in CHARLS 2011, 2013, 2015 and 2018 for the present study.

**Figure 2 jcm-11-06859-f002:**
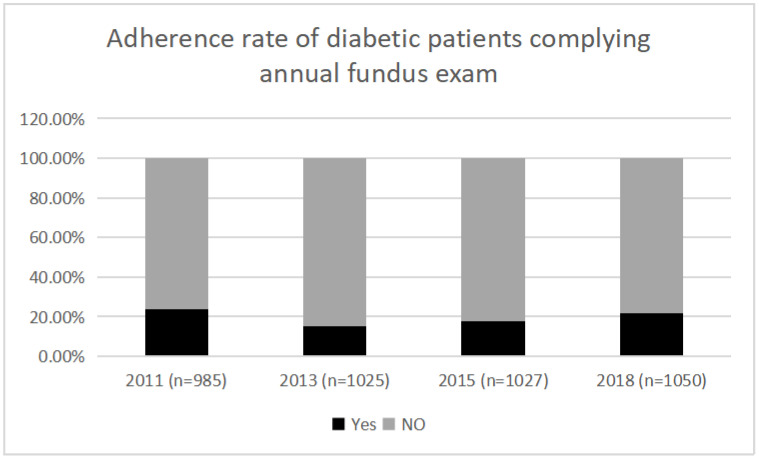
Adherence rates of participants with diabetes in CHARLS from 2011 to 2018.

**Table 1 jcm-11-06859-t001:** Descriptive characteristics of study sample from 2011 to 2018.

	2011 (*n* = 985)			2013 (*n* = 1025)			2015 (*n* = 1027)			2018 (*n* = 1050)		
Variables	Adherent (*n* = 231)	Non-Adherent (*n* = 754)	*p* Value	Adherent (*n* = 157)	Non-Adherent (*n* = 868)	*p* Value	Adherent (*n* = 180)	Non-Adherent (*n* = 847)	*p* Value	Adherent (*n* = 226)	Non-Adherent (*n* = 824)	*p* Value
Gender			0.1285			0.1364			0.3880			0.2867
Male	113 (48.9%)	326 (43.2%)		79 (50.3%)	381 (43.9%)		74 (41.1%)	378 (44.6%)		108 (47.8%)	361 (43.8%)	
Female	118 (51.1%)	428 (56.8%)		78 (49.7%)	487 (56.1%)		106 (58.9%)	469 (55.4%)		118 (52.2%)	463 (56.2%)	
Age	61.60 ± 9.52	61.07 ± 9.24	0.4477	62.31 ± 8.79	61.98 ± 9.27	0.6845	63.72 ± 8.67	63.68 ± 9.10	0.9555	63.67 ± 9.52	62.61 ± 9.39	0.1351
Education			<0.005			<0.005			<0.005			0.0047
Illiterate	35 (15.2%)	197 (26.1%)		24 (15.3%)	207 (23.8%)		32 (17.8%)	208 (24.6%)		46 (20.4%)	219 (26.6%)	
Less than elementary school	85 (36.8%)	304 (40.3%)		43 (27.4%)	371 (42.7%)		56 (31.1%)	356 (42.0%)		87 (38.5%)	348 (42.2%)	
Middle school	53 (22.9%)	146 (19.4%)		40 (25.5%)	168 (19.4%)		43 (23.9%)	166 (19.6%)		52 (23.0%)	150 (18.2%)	
High school or vocational school	39 (16.9%)	84 (11.1%)		36 (22.9%)	96 (11.1%)		33 (18.3%)	87 (10.3%)		27 (11.9%)	89 (10.8%)	
college and above	19 (8.2%)	23 (3.1%)		14 (8.9%)	26 (3.0%)		16 (8.9%)	30 (3.5%)		14 (6.2%)	18 (2.2%)	
Marital status			0.4756			0.1327			0.3252			0.3794
Yes	192 (83.1%)	611 (81.0%)		123 (78.3%)	723 (83.3%)		154 (85.6%)	699 (82.5%)		175 (77.4%)	660 (80.1%)	
No	39 (16.9%)	143 (19.0%)		34 (21.7%)	145 (16.7%)		26 (14.4%)	148 (17.5%)		51 (22.6%)	164 (19.9%)	
Smoke			0.9344			0.2357			0.3199			0.6137
Yes	80 (34.6%)	263 (34.9%)		67 (42.7%)	327 (37.7%)		68 (37.8%)	354 (41.8%)		76 (33.6%)	292 (35.4%)	
No	151 (65.4%)	490 (65.1%)		90 (57.3%)	541 (62.3%)		112 (62.2%)	493 (58.2%)		150 (66.4%)	532 (64.6%)	
Drinking status			0.432			0.2568			0.6172			0.4758
Drink more than once a month	42 (18.2%)	130 (17.3%)		38 (24.2%)	162 (18.7%)		33 (18.3%)	162 (19.1%)		52 (23.0%)	171 (20.8%)	
Drink but less than once a month	10 (4.3%)	50 (6.6%)		12 (7.6%)	64 (7.4%)		9 (5.0%)	58 (6.8%)		20 (8.8%)	59 (7.2%)	
None of these	179 (77.5%)	573 (76.1%)		107 (68.2%)	642 (74.0%)		138 (76.7%)	627 (74.0%)		154 (68.1%)	594 (72.1%)	
Using diabetes medication			<0.005			<0.005			<0.005			<0.005
Yes	190 (82.3%)	512 (67.9%)		145 (92.4%)	534 (61.5%)		156 (86.7%)	512 (60.4%)		167 (73.9%)	469 (56.9%)	
No	41 (17.7%)	242 (32.1%)		12 (7.6%)	334 (38.5%)		24 (13.3%)	335 (39.6%)		59 (26.1%)	355 (43.1%)	
Non-medication treatments			0.1086			<0.005			<0.005			<0.005
Yes	105 (45.5%)	298 (39.5%)		83 (52.9%)	335 (38.6%)		90 (50.0%)	298 (35.2%)		122 (54.0%)	340 (41.3%)	
No	126 (54.5%)	456 (60.5%)		74 (47.1%)	533 (61.4%)		90 (50.0%)	549 (64.8%)		104 (46.0%)	484 (58.7%)	
Multi-morbidities			0.9407			0.1144			0.5164			0.2161
Yes	199 (86.1%)	651 (86.3%)		141 (89.8%)	738 (85.0%)		163 (90.6%)	753 (88.9%)		188 (83.2%)	655 (79.5%)	
No	32 (13.9%)	103 (13.7%)		16 (10.2%)	130 (15.0%)		17 (9.4%)	94 (11.1%)		38 (16.8%)	169 (20.5%)	
Vision impairment			0.3551			0.2759			0.3432			0.4458
Yes	109 (47.2%)	382 (50.7%)		70 (44.6%)	428 (49.3%)		80 (44.4%)	344 (40.6%)		75 (33.2%)	296 (35.9%)	
No	122 (52.8%)	372 (49.3%)		87 (55.4%)	440 (50.7%)		100 (55.6%)	503 (59.4%)		151 (66.8%)	528 (64.1%)	
Regular physical exam			<0.005			<0.005			<0.005			<0.005
Yes	179 (77.5%)	496 (65.8%)		127 (80.9%)	479 (55.2%)		137 (76.1%)	449 (53.0%)		172 (76.1%)	458 (55.6%)	
No	52 (22.5%)	258 (34.2%)		30 (19.1%)	389 (44.8%)		43 (23.9%)	398 (47.0%)		54 (23.9%)	366 (44.4%)	
Insurance coverage			0.4885			0.3779			0.0142			0.6999
Yes	220 (95.2%)	709 (94.0%)		151 (96.2%)	820 (94.5%)		166 (92.2%)	816 (96.3%)		222 (98.2%)	806 (97.8%)	
No	11 (4.8%)	45 (6.0%)		6 (3.8%)	48 (5.5%)		14 (7.8%)	31 (3.7%)		4 (1.8%)	18 (2.2%)	

**Table 2 jcm-11-06859-t002:** Univariate logistic regression of factors associated with adherence to annual fundus exam.

Variables	2011 (*n* = 985)	2013 (*n* = 1025)	2015 (*n* = 1027)	2018 (*n* = 1050)
	Odd ratio (95% Confidence Interval)			
Gender				
Female	Reference			
Male	1.254 (0.933, 1.686)	1.295 (0.921, 1.820)	0.866 (0.625, 1.200)	1.174 (0.874, 1.577)
Age	1.006 (0.991, 1.022)	1.004 (0.985, 1.023)	1.001 (0.983, 1.019)	1.012 (0.996, 1.028)
Education				
Illiterate	Reference			
Less than elementary school	1.566 (1.016, 2.413) *	1.000 (0.590, 1.694)	1.022 (0.641, 1.631)	1.190 (0.802, 1.767)
Middle school	2.033 (1.261, 3.278) **	2.054 (1.190, 3.543) *	1.684 (1.020, 2.779) *	1.650 (1.055, 2.583) *
High school or vocational school	2.600 (1.541, 4.387) ***	3.234 (1.828, 5.722) ***	2.466 (1.427, 4.260) **	1.444 (0.846, 2.467)
college and above	4.626 (2.283, 9.373) ***	4.644 (2.140, 10.08) ***	3.467 (1.701, 7.064) **	3.703 (1.719, 7.976) **
Marital status				
No	Reference			
Yes	1.144 (0.775, 1.689)	0.725 (0.477, 1.104)	1.254 (0.798, 1.970)	0.853 (0.597, 1.217)
Smoke				
No	Reference			
Yes	0.987 (0.724, 1.345)	1.232 (0.872, 1.739)	0.846 (0.607, 1.177)	0.923 (0.677, 1.259)
Drinking status				
None of these	Reference			
Drink but less than once a month	0.641 (0.318, 1.289)	1.125 (0.588, 2.154)	0.705 (0.341, 1.457)	1.308 (0.764, 2.238)
Drink more than once a month	1.034 (0.703, 1.522)	1.408 (0.936, 2.117)	0.926 (0.610, 1.405)	1.173 (0.820, 1.677)
Using diabetes medication				
No	Reference			
Yes	2.195 (1.515, 3.179) ***	7.557 (4.129, 13.83) ***	4.253 (2.708, 6.678) ***	2.142 (1.544, 2.972) ***
Non-medication treatments				
No	Reference			
Yes	1.279 (0.950, 1.723)	1.785 (1.268, 2.512) **	1.842 (1.332, 2.548) ***	1.670 (1.242, 2.245) **
Multi-morbidities				
No	Reference			
Yes	0.985 (0.643, 1.511)	1.552 (0.896, 2.690)	1.197 (0.695, 2.061)	1.276 (0.866, 1.880)
Vision impairment				
No	Reference			
Yes	0.872 (0.649, 1.172)	0.827 (0.588, 1.164)	1.170 (0.846, 1.618)	0.886 (0.649, 1.210)
Regular physical exam				
No	Reference			
Yes	1.784 (1.265, 2.515) **	3.437 (2.259, 5.230) ***	2.824 (1.954, 4.081) ***	2.545 (1.820, 3.559) ***
Insurance coverage				
No	Reference			
Yes	1.271 (0.646, 2.500)	1.472 (0.619, 3.500)	0.450 (0.234, 0.865)	1.239 (0.415, 3.699)

*p* < 0.05 *****; *p* < 0.005 ******; *p* < 0.0005 *******.

**Table 3 jcm-11-06859-t003:** Multivariate logistic regression of factors associated with adherence to annual fundus exams.

Variables	2011 (*n* = 985)	2013 (*n* = 1025)	2015 (*n* = 1027)	2018 (*n* = 1050)
	Odd ratio (95% Confidence Interval)			
Education				
Illiterate	Reference			
Less than elementary school	1.490 (0.961, 2.309)	0.911 (0.527, 1.577)	0.885 (0.545, 1.435)	1.184 (0.789, 1.777)
Middle school	1.862 (1.147, 3.021) *	1.571 (0.888, 2.780)	1.397 (0.829, 2.354)	1.446 (0.910, 2.298)
High school or vocational school	2.488 (1.464, 4.227) **	2.605 (1.422, 4.771) **	1.792 (1.011, 3.175) *	1.303 (0.752, 2.259)
College and above	4.067 (1.977, 8.369) ***	3.208 (1.399, 7.359) *	2.380 (1.116, 5.075) *	1.303 (0.752, 2.259) *
Using diabetes medication				
No	Reference			
Yes	2.113 (1.448, 3.084) ***	7.124 (3.856, 13.16) ***	3.772 (2.379, 5.979) ***	2.083 (1.487, 2.919) ***
Non-medication treatments				
No	Reference			
Yes	1.310 (0.906, 1.893)	1.310 (0.906, 1.893)	1.515 (1.076, 2.132) *	1.521 (1.118, 2.070) *
Regular physical exam				
No	Reference			
Yes	1.665 (1.171, 2.366) **	2.917 (1.881, 4.524) ***	2.408 (1.645, 3.525) ***	2.410 (1.704, 3.407) ***

*p* < 0.05 *****; *p* < 0.005 ******; *p* < 0.0005 *******.

## Data Availability

The original dataset of CHARLS is available on http://charls.pku.edu.cn/ (accessed on 3 December 2013).

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
