# Peer review of "Adherence to Annual Fundus Exams among Chinese Population with Diagnosed Diabetes"

_jcm, 2022, doi:10.3390/jcm11226859_

Round 1

Reviewer 1 Report

Zhou and colleagues utilize CHARLS to explore trends in annual eye exam adherence in patients with diabetes across several years, as well as associations with poor adherence, in China. The study adds to the growing epidemiology literature of patient adherence in diabetic eye exams. The figures and tables are very helpful, especially the p-values in Table 1 for each year that clearly demonstrates the statistical findings of the study. The authors should be commended in performing this longitudinal retrospective study, but the paper has several key areas of improvement.

Overall/Major Comments

Which variables were included in the multivariate analysis? Traditionally, variables with p-values <0.1 are included in the multivariate analysis. It is unclear which variables met this criterion, so it is unclear which variables were controlled in the multivariate analysis. The authors should be more specific in the Methods section and potentially re-run the multivariate analysis. This is because inclusion of a variable showing p>0.1 in the univariate analysis into the multivariate analysis would change the odds ratios reported.

The discussion is very long, has subheadings, and distracts the reader and needs to be shortened. More specifically, the first section before the “Educational level, diabetic medication, non-medication treatments and physical exam section” can be shortened to 2 paragraphs, citing the most salient points. The section on the educational level and other parameters can also be shortened to 1-2 paragraphs. There is significant focus on educational level and attainment does not add great value and goes beyond the scope of fundus examinations (eg. vitrectomy etc.). Salient points can likely be summarized in 1 paragraph- most of which are summarized by the first sentence of each paragraph currently. The strengths and limitations section can also be cut down to one paragraph to only include pertinent strengths in 2-3 sentences and limitations in 2-3 sentences. Subheadings can then be omitted in the discussion section.

The language should be in third person and not use “we” or “I”; outlined some changes in the minor comments section. “Diabetic patient” should also be changed to “patient with diabetes” for inclusive and non-disease-defining language use.

Minor Comments

Line 45- “some” is redundant

Line 49- “covering” should be “coverage”

Line 51- “actually” is redundant

Lines 55-56: Can be rephrased as “significantly low adherence rates in developing countries are even more worrisome.”

Line 133- “We noticed that” is redundant

Line 135- “less” should be “fewer” in diabetes medication use

Lines 157-159: Redundant and can be omitted to start with the fact that China has the largest number of patients with diabetes and that there have been research looking into screening programs among Chinese patients with diabetes.

Line 171- “State” should be “States”

Tables- Insurance “cover” should be “coverage”

Author Response

Zhou and colleagues utilize CHARLS to explore trends in annual eye exam adherence in patients with diabetes across several years, as well as associations with poor adherence, in China. The study adds to the growing epidemiology literature of patient adherence in diabetic eye exams. The figures and tables are very helpful, especially the p-values in Table 1 for each year that clearly demonstrates the statistical findings of the study. The authors should be commended in performing this longitudinal retrospective study, but the paper has several key areas of improvement.

Overall/Major Comments

Which variables were included in the multivariate analysis? Traditionally, variables with p-values <0.1 are included in the multivariate analysis. It is unclear which variables met this criterion, so it is unclear which variables were controlled in the multivariate analysis. The authors should be more specific in the Methods section and potentially re-run the multivariate analysis. This is because inclusion of a variable showing p>0.1 in the univariate analysis into the multivariate analysis would change the odds ratios reported.

Response to reviewer:

In the original manuscript, we stated in the statistical methods section that, “Multivariable logistic regressions were then performed using significant results of the above listed demographic and clinical variables from univariate analysis to determine which variables were independently associated with improved adherence to annual fundus exams.”. Actually, we did adopt the inclusion criteria as the reviewer mentioned: “Traditionally, variables with p-values <0.1 are included in the multivariate analysis.”. Thus, we have amended this statement in the revised version as: "Multivariable logistic regressions were then performed using filtered variables (with p values<0.1) from univariate analysis to determine which variables were independently associated with improved adherence to annual fundus exams."

Additionally, we added age and gender in the multivariate analysis in the original manuscript. It is widely appreciated in numerous studies that these two variables are essential and fundamental characteristics of patients which have comprehensive influences and broad interactions with other factors. Thus, it might make some sense to preserve these two variables in the multivariate models. On the other hand, the p values of age and gender according to univariate analyses were indeed >0.1 according to our univariate analysis, and we agreed with the reviewer that “inclusion of variables with p value>0.1 in the univariate analysis into the multivariate analysis would change the ORs.”

Since both inclusion logics have certain reasons for adoption.

We re-runed the multivariate analysis with only significant variables according to univariate analyses (without age and gender), submitted as New Table 3. Please see the attachment

If we compare the Original Table 3 in the revised manuscript, and the New Table 3 submitted separately with revised manuscript, the results (ORs) and their significances are very much similar.

Therefore, from our point of view, both statistical logics might be qualified to perform our work. If the reviewer prefers the New Table 3, then we will replace the Original one.

We hope that our explanations and re-runed work could get your appreciation. Thank you.

The discussion is very long, has subheadings, and distracts the reader and needs to be shortened. More specifically, the first section before the “Educational level, diabetic medication, non-medication treatments and physical exam section” can be shortened to 2 paragraphs, citing the most salient points. The section on the educational level and other parameters can also be shortened to 1-2 paragraphs. There is significant focus on educational level and attainment does not add great value and goes beyond the scope of fundus examinations (eg. vitrectomy etc.). Salient points can likely be summarized in 1 paragraph- most of which are summarized by the first sentence of each paragraph currently. The strengths and limitations section can also be cut down to one paragraph to only include pertinent strengths in 2-3 sentences and limitations in 2-3 sentences. Subheadings can then be omitted in the discussion section.

Response to reviewer: You have our most sincere gratitude for this excellent suggestion! Some of our findings are quite novel and interesting, which were inviting for us to make expatiatory discussions in the original manuscript. During this revision, we carefully reviewed our discussion section and we are totally agreed with the reviewer that this section is quite tedious. We have condensed the whole content of the discussion section and strengths & limitations part accordingly. All subheadings have been omitted in the revised version. We feel that the discussion section now is much more clear, coherent and concise. Thus, heartfelt thank you to our reviewer!

The language should be in third person and not use “we” or “I”; outlined some changes in the minor comments section. “Diabetic patient” should also be changed to “patient with diabetes” for inclusive and non-disease-defining language use.

Response to reviewer: We have carefully reviewed our manuscript and improved its expression (tried best to reduce frequency of using “we” or “I”). And thank you so much for careful and thoughtful suggestions in the minor comments section. We have made amendments accordingly in the revised version. And we apologized for reckless usage of “diabetic patient” in our original manuscript. It is noteworthy to use inclusive and non-disease-defining expressions. We have changed all “diabetic patient” into “patient with diabetes”, and “diabetic population” into “population with diabetes”. Thank you very much for these suggestions!

Minor Comments:

Line 45- “some” is redundant

Line 49- “covering” should be “coverage”

Line 51- “actually” is redundant

Lines 55-56: Can be rephrased as “significantly low adherence rates in developing countries are even more worrisome.”

Line 133- “We noticed that” is redundant

Line 135- “less” should be “fewer” in diabetes medication use

Lines 157-159: Redundant and can be omitted to start with the fact that China has the largest number of patients with diabetes and that there have been research looking into screening programs among Chinese patients with diabetes.

Line 171- “State” should be “States”

Tables- Insurance “cover” should be “coverage”

Reviewer 2 Report

The manuscript entitled “Adherence to annual fundus exams among Chinese population with diagnosed diabetes, 2011-2018” describes the adherence rate to annual fundus examination among Chinese diabetic population across 4 waves (2011 – 2018) revealing a low adherence to clinical guideline for eye health.

The manuscript is well written and easy to understand. The data collected in the manuscript is very extensive both in patients and years which provides a general insight of the adherence status of the Chinese diabetic population to the fundus exam. The references used in the manuscript are recent and adequate.  Regarding the novelty of the manuscript, as far as I am concerned this is the first time that the adherence rate to annual fundus examination among Chinese diabetic population. The authors take into account the strengths and limitations of the study.

In my opinion, the results shown in the present manuscript are interesting for a broader community, nonetheless. Despite its great potential, the authors should revise some small issues to improve the readability of the manuscript.

·        I would rework Table 1, to improve its readability.

·        When writing p values, I would suggest being uniform (it is written as p and P) in usage and use italics (p value).

·        Some reference numbers are attached to the previous word, while others are not. Following the authors guidelines leave a blank space between the previous word and the reference number.

Best regards

Author Response

The manuscript entitled “Adherence to annual fundus exams among Chinese population with diagnosed diabetes, 2011-2018” describes the adherence rate to annual fundus examination among Chinese diabetic population across 4 waves (2011 – 2018) revealing a low adherence to clinical guideline for eye health.

The manuscript is well written and easy to understand. The data collected in the manuscript is very extensive both in patients and years which provides a general insight of the adherence status of the Chinese diabetic population to the fundus exam. The references used in the manuscript are recent and adequate. Regarding the novelty of the manuscript, as far as I am concerned this is the first time that the adherence rate to annual fundus examination among Chinese diabetic population. The authors take into account the strengths and limitations of the study.

In my opinion, the results shown in the present manuscript are interesting for a broader community, nonetheless. Despite its great potential, the authors should revise some small issues to improve the readability of the manuscript.

  • I would rework Table 1, to improve its readability.

Response to reviewer: Thanks for this suggestion. We have now bolded all the p values of all the significant results to highlight and reformatted the variables rank to improve its readability as well.

  • When writing p values, I would suggest being uniform (it is written as p and P) in usage and use italics (p value).

Response to reviewer: All the expressions of "p values" have been amended as you suggested. Thank you very much.

  • Some reference numbers are attached to the previous word, while others are not. Following the authors guidelines leave a blank space between the previous word and the reference number.

Response to reviewer: We have carefully reviewed the manuscript and make amendments as you suggested. Thank you very much.

Best regards

Round 2

Reviewer 1 Report

The authors have made great changes to the manuscript based on feedback. The new Table with the new analysis would be preferred given its robust statistical analysis. The only change I recommend is the deletion of the "Strengths and Limitations" wording (it is a bolded subheading) in the discussion section, as these paragraphs seemlesly integrate with the remainder of the discussion anyway.

Author Response

We are very much grateful to your suggestion on multivariate analysis and we have replaced the Original Table 3 with the New Table 3 in the revised version. Also, the "Strengths and Limitations" wording has been deleted. Thank you!